# Extended Aerosol Optical Depth (AOD) time series analysis in an Alpine Valley: A Comparative Study from 2007 to 2023

Jochen Wagner[1], Alma Anna Ubele[1], Verena Schenzinger[1], and Axel Kreuter[1, 2]

[1]Institute of Biomedical Physics, Medical University of Innsbruck,Müllerstraße 44, 6020 Innsbruck, Austria
[2]LuftBlick Earth Observation Technologies, Fritz-Konzert-Strasse 4, 6020 Innsbruck, Austria

**Correspondence:** Jochen Wagner (jochen.wagner@i-med.ac.at)

**Abstract.** This study presents an extended analysis of aerosol optical depth at 501 nm (AOD) in the Alpine valley of Innsbruck, Austria, from 2007 to 2023, and offers a comparative analysis with the Alpine station of Davos, Switzerland. AOD is derived from ground-based sunphotometer measurements of direct spectral irradiance during daytime. The Davos Station is part of the AErosol Robotic NETwork (AERONET), a global network providing high quality, ground based remote sensing aerosol data and complies with the relevant requirements. The Innsbruck station does not belong to AERONET, but the AOD retrieval algorithm is very similar. Building upon previous research conducted until 2012, the presented study aims to provide a comprehensive understanding of the long-term trends and seasonal variations in aerosol characteristics in Central Alpine regions. We observed the typical mid latitude annual cycle with a maximum in July and a minimum in December. The AOD trends per decade for both stations are declining, -27.9 x $10^{-3}$ for Innsbruck and -9.9 x $10^{-3}$ for Davos.

## 1 Introduction

The interplay between atmospheric aerosols and environmental dynamics has long been a subject of keen scientific interest, particularly in the context of climate change (Li et al. (2022)), air quality, cloud microphysics (Tiwari et al. (2023)) and ecological impacts (Zhou et al. (2021)). Aerosol Optical Depth (AOD) is a pivotal parameter in this domain, offering a quantifiable measure of aerosol concentration in the Earth's atmosphere. It quantifies the cumulative effect of aerosol scattering and absorption along the path of sunlight through the atmosphere. AOD is unitless and provides an indication of atmospheric clarity, essential for climatological and environmental research. The primary method for determining AOD is through the use of sun photometers, which measure the direct solar irradiance reaching the Earth's surface. The basic principle behind these measurements is the Lambert-Beer law, a fundamental equation that relates the intensity of light to the properties of the material through which it is passing.

$$I = I_0(R) \cdot e^{-\tau(\lambda) \cdot m} \tag{1}$$

with

- $I$ is the observed intensity of sunlight after passing through the atmosphere

- $I_0(R)$ is the original intensity of sunlight before entering the Earth's atmosphere, dependent on the sun-earth distance $R$

- $\tau(\lambda)$ is the optical depth at wavelength $\lambda$, which includes contributions from aerosols, gases, and other atmospheric constituents

- $m$ is the optical air mass, a factor that accounts for the path length through the atmosphere, which depends on the solar zenith angle ($m \sim cos(sza)$)

A detailed description of the retrieval of AOD from sunphotometer measurements in Innsbruck is given in Wuttke et al. (2012) and in Sinyuk et al. (2020) for the AERONET AOD retrieval respectively.

Satellite derived AOD with global coverage improves our knowledge on the distribution (Levy et al. (2009)). However, satellite retrievals face limitations due to their viewing geometry, where light traverses the atmosphere twice and reflects off the Earth's surface, complicating accurate measurement, whereas ground based remote sensing observations meet the World Meteorological Organization (WMO) traceability requirements in more than 95% of the measurements (Cuevas et al. (2019)) and allow robust trend analyses (Kazadzis et al. (2018)). High quality AOD time series are of special importance regarding climate observations (Kassianov et al. (2021)). This study aims to deepen our understanding of aerosol behavior in the Alpine valleys of Innsbruck, Austria, and Davos, Switzerland. Unfortunately, other stations from AERONET (Giles et al. (2019)) like Zugspitze and Bolzano have only very limited measurement series.

The Alpine region, characterized by its distinct topography and climatic conditions, presents a natural laboratory for studying aerosols (Ingold et al. (2001)). The complex interactions of local and regional meteorological patterns, coupled with anthropogenic influences, make this area particularly interesting for long-term environmental observations of aerosols (Lenoble et al. (2008)). In this context, the city of Innsbruck, a valley station in the centre of the Tyrolean Alps, and the high-altitude station of Davos in Switzerland, provide contrasting yet complementary settings for examining aerosol characteristics.

Innsbruck, situated in the broad Inn Valley, is a prominent cultural and academic center in western Austria with about 132,000 residents. The city's geographical position in a large valley facilitates unique meteorological conditions, characterized by pronounced seasonal variations. Typical weather patterns include relatively dry winters and wetter summers, with occasional föhn winds influencing both temperature and precipitation levels. Davos, on the other hand, is a high-altitude town located in the Swiss Alps. It has a smaller population of about 11,000 inhabitants, which can swell significantly during tourist seasons. Davos experiences a subarctic climate, which includes long, snow-rich winters and cool summers. The meteorological setup in Davos leads to a distinct aerosol composition primarily influenced by tourism-related activities and seasonal sports events, contrasting with Innsbruck's more urban aerosol sources from vehicular traffic and industrial emissions. Both sites, therefore, offer contrasting environments for the study of aerosols, significantly enhancing the comparative analysis of long-term AOD trends.

In Europe, strict environmental regulations and implementations of cleaner technologies since the late 20th century have significantly reduced aerosol emissions, while an upward trend has been observed in other regions (Yu et al. (2020)). This "global brightening" effect became evident since the 1980s. It seems, that this effect is still ongoing since many studies show a decreasing AOD in Europe over the last 20 years (Cherian and Quaas (2020)). Our research is anchored in the long history of

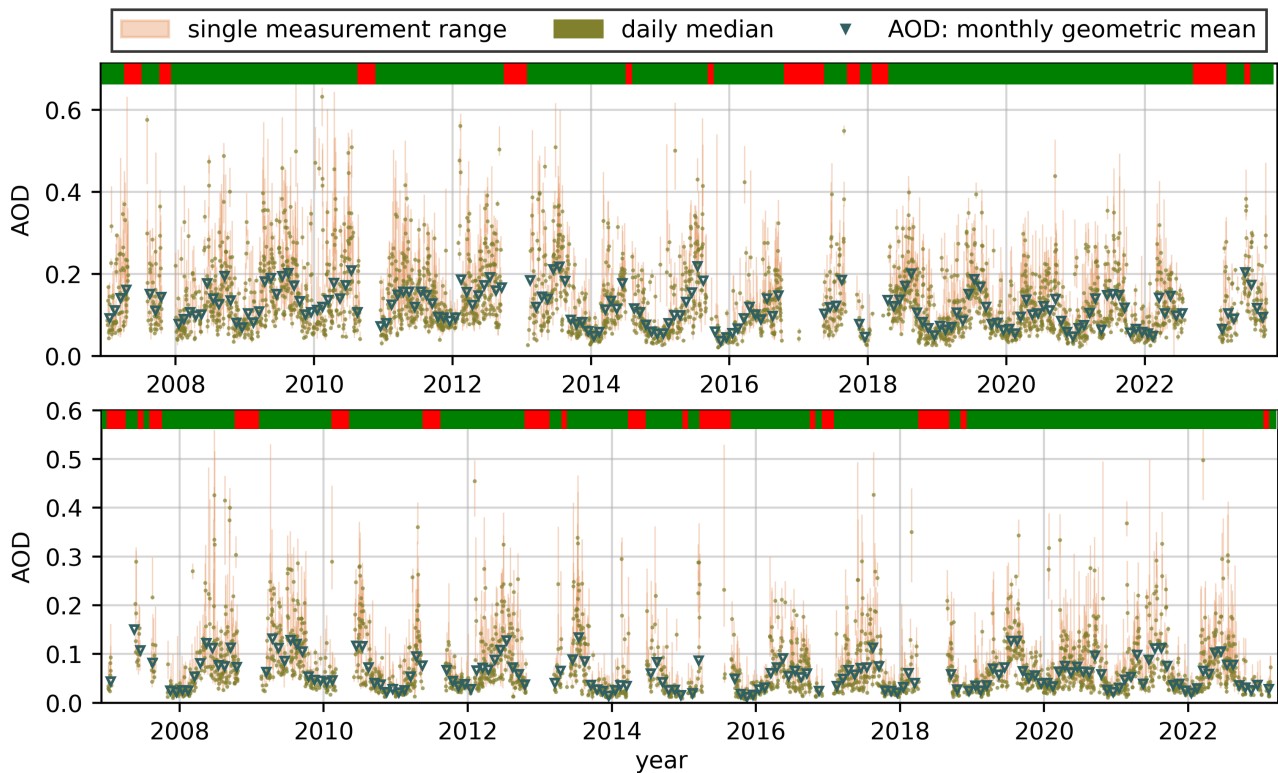

**Figure 1.** AOD time series from Innsbruck (top) and Davos (bottom). Individual measurements (Innsbruck - minute intervals; Davos - 10 minutes intervals) are shown as pink dots, daily values as greenish circles and monthly averages (geometric mean) as blueish triangles. The data availability of the monthly averages is shown at the top op each graph.

aerosol studies in Alpine environments, notably extending the work of Wuttke et al. (2012) and drawing comparative insights from recent findings by Karanikolas et al. (2022). By analyzing a 17-year AOD dataset, this study seeks to uncover the long-term trends and seasonal variabilities of aerosols in two Alpine valleys. The extended timeframe of our analysis, spanning from 2007 to 2023, allows for a detailed exploration of the temporal evolution of aerosol characteristics, contributing to a broader understanding of their role in regional and global climatic systems.

The significance of this study lies not only in its extended temporal scope but also in its contribution to the ongoing discourse on environmental and climatic changes. By examining the trends and patterns in AOD data, we aim to provide valuable insights into the underlying processes driving aerosol distribution and concentration in the Alpine region. This research holds valuable information for future environmental policies and strategies aimed at mitigating the impacts of atmospheric aerosols on climate, ecosystems, and human health.

**Table 1.** The number of measurements of the datasets, the time period used and the number of days and months considered as valid with the percentage of valid days/months in brackets.

| Station | Lat | Lon | Elevation | Period | N | Valid Days | Valid Months |
|---|---|---|---|---|---|---|---|
| Innsbruck | 47.26417 ° N | 11.38569 ° E | 620 m | 01/2007 - 10/2023 | 612962 | 2973/6117 (48.6%) | 168/202 (83.2%) |
| Davos | 46.81281° N | 9.84369 ° E | 1589 m | 01/2007 - 02/2023 | 78124 | 2479/5893 (42.1%) | 154/194 (79.4%) |

## 2 Methods

Precision Filter Radiometers (PFRs) are engineered to assess background aerosol conditions and have participated in sun photometer intercomparisons, like the CIMEL devices used in the global AERONET network to ensure data quality assurance. The discrepancies between PFRs and CIMEL devices used in the global AERONET network (Holben et al. (2001)), consistently fall within a +/- 0.01 AOD range. The Innsbruck PFR performed even better during the intercomparison campaign in Davos in October 2021 (Kazadzis et al. (2023)). Long-term analyses confirm the excellent traceability of AERONET AOD measurements to the World AOD standard at 500 nm (Cuevas et al. (2019)). Furthermore, regular calibrations of the PFR in Innsbruck, conducted in Davos, have shown remarkably stable calibration coefficients for the 501 nm channel over the past 17 years, with relative changes ranging from -0.5% to +0.7%.

Utilizing a robust dataset collected over 17 years in Innsbruck and Davos (see Figure 1), we employ best practices (Sayer and Knobelspiesse (2019), Weatherhead et al. (1998)) to analyse the AOD time series, focusing on identifying trends, patterns, and anomalies. Both time series start in January 2007. The time series of Innsbruck ends in October 2023, whereas data from Davos were only available until February 2023. The temporal resolution in Innsbruck is 1 min and in Davos 10 min. Furthermore the data availability with 48.6% /42.1% daily and 82.2%/79.4% monthly (see table 1), for Innsbruck and Davos respectively, is also very similar and remarkably high, given, that measurements are only possible when the sun is above the horizon and not obscured by clouds.

We calculated daily median values only for days with at least three measurements (also standard in AERONET processing). The daily AOD climatology was derived by calculating the median for each day of the year (see Figure 3 and 4). From these values the monthly geometric mean was calculated if there were at least five valid days available. With this approach we calculated the monthly AOD from 168 out of 202 months (83.2%) in Innsbruck and 154 out of 194 months (79.4%) in Davos (table 1) . The missing data, accounting for approximately 20 % of the total dataset, are not uniformly distributed throughout the year. Our analysis indicates that these gaps are more prevalent during the winter months, primarily due to shorter daylight hours. The primary reasons for these data gaps are twofold: instrument calibration and failures. Calibration periods are scheduled routinely to ensure the accuracy and reliability of our measurements but result in temporary interruption of data collection.

The study also deals with a comparative analysis, highlighting the similarities and differences in aerosol behavior between the two locations. One of the main aims of the work is to perform a trend analysis on the monthly time series. First, we

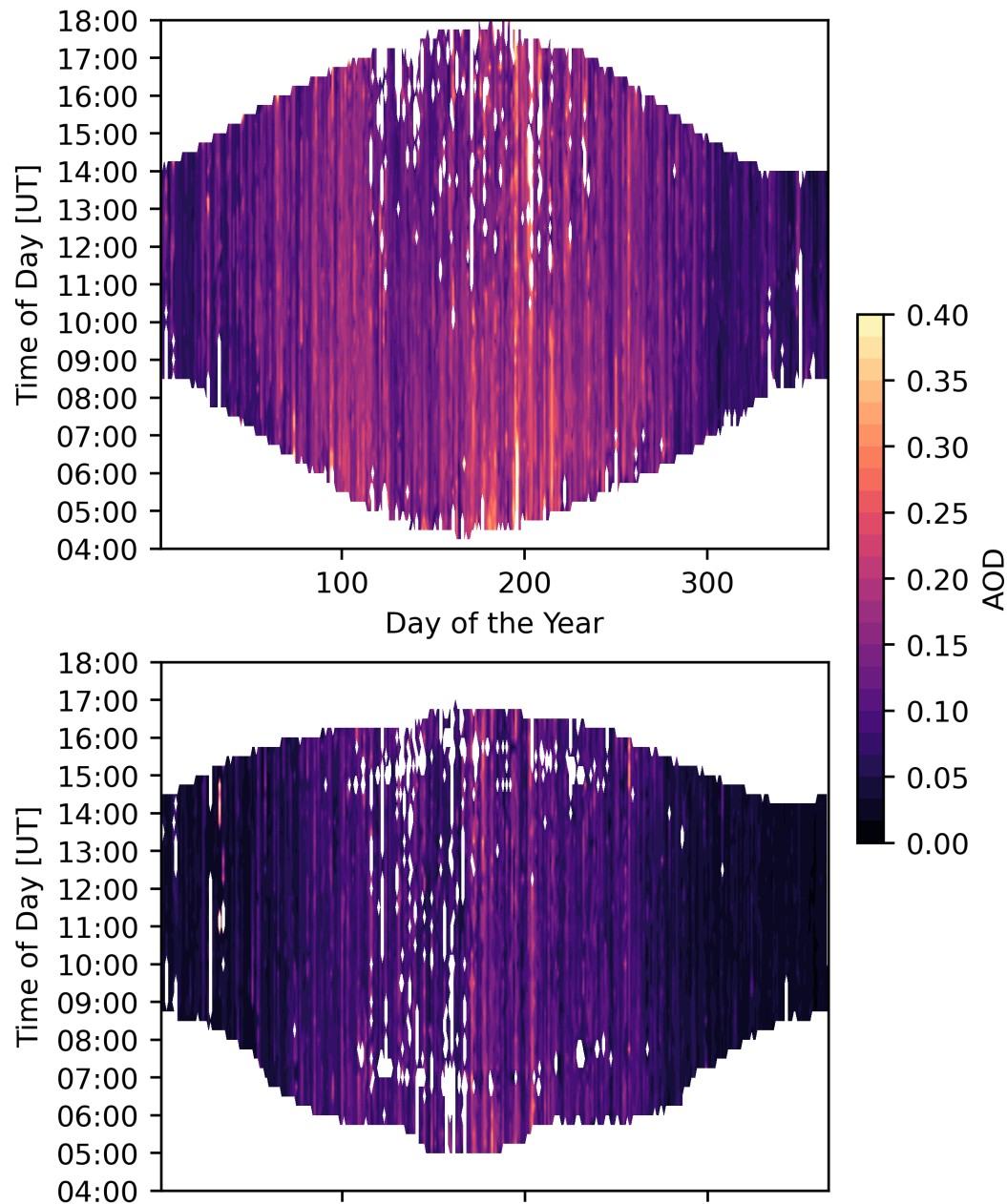

**Figure 2.** Median AOD for each 15 min interval on each day of the year in Innsbruck (top) and Davos (bottom). White areas indicate that there are no data available at these time points in the 17-year time series.

deseasonalized the time series of the monthly AOD and applied linear fitting on the residuals. Additionally we calculated the trends for each month using ideally 17 values. Our findings reveal negative trends in AOD.

95

## 2.1 Results

A closer look at the two time series (Figures 1) reveals the typical lognomal distribution of the AOD measurements (O'Neill et al. (2000)). The highest value of 0.632 was measured in Innsbruck on 12 February 2010 and in Davos (0.864) on 2 February 2012 (both values outside the displayed y-range). The lowest value was observed on 2 November 2015 in Innsbruck (0.021) and on 7 November 2015 in Davos (0.007). Longer data gaps occur in both time series due to device failures or calibrations. Short gaps result from periods of bad weather. The typical annual variation is already recognizable, especially in the monthly averages.

The AOD is derived by measuring the direct irradiance of the sun. Therefore measurement errors often correlate with the zenith and azimuth angle. Figure 2 provide a good visual overview of the average annual and daily variation of the AOD at the two locations; no clear diurnal variation can be observed at either location. In Innsbruck, it is noticeable that there are many data gaps in the afternoon, especially in summer, which is probably due to convective clouds. In Davos, data gaps occur mainly in spring. This effect might occur due to more convective clouds in the afternoon during and after the melting period in spring and early summer. In addition, there are particularly many data gaps here in the summer half-year with a solar zenith angle of approx. 15 degrees both in the morning and in the evening. This effect is very likely due to skyscans (almucantar and principle plain) mandatory for AERONET stations.

Due to the short time series (30 years is the standard for climatologies) and the data gaps due to cloudy days, the climatologies of the two stations on a daily basis (Figures 3) show (still) strong fluctuations. Nevertheless, the representation offers added value because the lognormal distribution becomes clear and extreme events can be quickly identified.

The climatologies of the two stations on a monthly basis are a central result of this study. The annual mean value (geometric mean of the daily values) is 0.115 in Innsbruck and 0.054 in Davos. The month with the highest AOD is July (0.163/0.093) and the month with the lowest AOD is December (0.062/0.025) for Innsbruck and Davos respectively. The standard error correlates with the absolute values. This behaviour is typical for lognormal distributed data.The different altitudes and increased influence of human activities apparently only have an influence on the absolute value of the AOD, but not on the characteristic diurnal variation. The month of May is an exception. Here there is a local minimum in Innsbruck, while the month is unremarkable in Davos. This effect might be caused by differences in the annual cycle of the biosphere due to the difference in altitude between Davos and Innsbruck. However, further investigations are needed to prove this hypothesis. Especially continuous lidar observations of aerosol extinction profiles could provide a clearer distinction between boundary layer and free tropospheric aerosols.

We calculated the trend from the deseasonalized monthly AOD time series (Figure 5). For both stations a declining trend is obvious. However, we observed only a weak (Davos) to moderate (Innsbruck) correlation. For Innsbruck we calculated a trend of -27.9 x $10^{-3}$ with $p = 0.00$ and $r = -0.45$ and for Davos -9.9 x $10^{-3}$ with $p = 0.00$ and $r = -0.24$. These trends are in line with the findings of Yang et al. (2020) and Wei et al. (2019). Additionally we calculated the AOD trends per decade also for each month (table 2). The monthly trend calculations, due to the limited number of data points (11 - 16), are not yet very meaningful. The requirements for significance ($p < 0.05$ and $|r| > 0.6$) are only met for a few months (January, October

**Table 2.** AOD trends per decade x $10^{-3}$ for each month in Innsbruck and Davos - number of valid months N in brackets. Bold numbers indicate significant trends.

| Month | Innsbruck trend (N) | p - value | r | Davos trend (N) | p - value | r |
|---|---|---|---|---|---|---|
| 1 | **-25.9** (13) | 0.01 | -0.65 | -2.6 (12) | 0.60 | -0.16 |
| 2 | -43.6 (15) | 0.04 | -0.53 | -1.9 (14) | 0.82 | -0.07 |
| 3 | -18.9 (15) | 0.12 | -0.41 | -2.3 (14) | 0.81 | -0.07 |
| 4 | -29.5 (16) | 0.03 | -0.54 | -18.1 (11) | 0.19 | -0.43 |
| 5 | -38.6 (15) | 0.04 | -0.54 | **-43.8** (11) | 0.01 | -0.73 |
| 6 | -21.3 (16) | 0.29 | -0.28 | -18.7 (12) | 0.09 | -0.51 |
| 7 | -31.4 (15) | 0.11 | -0.42 | -17.5 (12) | 0.37 | -0.28 |
| 8 | -6.1 (16) | 0.76 | -0.08 | 6.4 (13) | 0.64 | 0.14 |
| 9 | -29.0 (13) | 0.10 | -0.47 | -12.7 (15) | 0.40 | -0.23 |
| 10 | **-50.7** (11) | 0.00 | -0.82 | -10.5 (14) | 0.27 | -0.31 |
| 11 | -20.2 (11) | 0.12 | -0.49 | 3.1 (13) | 0.67 | 0.13 |
| 12 | **-27.6** (12) | 0.02 | -0.68 | -0.6 (13) | 0.91 | -0.03 |
| all | -27.9 (168) | 0.00 | -0.45 | -9.9 (154) | 0.00 | -0.24 |

and December in Innsbruck and May in Davos). Nevertheless, a fairly consistent pattern emerges again. With the exception of August and September in Davos all trends are negative. May shows the strongest negative trend in Davos and the third strongest negative trend in Innsbruck. August is the month with the least decrease in Innsbruck, or even a slight increase in Davos. In contrast, there are strong trend differences between Innsbruck and Davos in October and February.

## 3 Conclusions and Outlook

Overall, the results in AOD statistics for Innsbruck and Davos are remarkably consistent. The trends are as expected (Yang et al. (2020) and Wei et al. (2019)) and show, that the decline of AOD in the last 17 years can be observed in the lower and also the upper atmosphere. The observed decline is very likely due to a decline of anthropogenic emissions (Myhre et al. (2017)). It seems, that the local minimum in May in Innsbruck is becoming even more pronounced. For a better understanding of the aerosol behaviour, it is essential to distinguish between boundary layer aerosols and aerosols in the free troposphere and additionally investigations on local emissions and land use changes are worthwhile.

In summary, this study represents a significant step forward in our comprehension of aerosol climatology in the Alpine region, offering a nuanced understanding of the environmental statistics and long-term trends of aerosols in Innsbruck and Davos.

*Data availability.* The AOD measurements from Davos is available via Aeronet: https://aeronet.gsfc.nasa.gov/new_web/photo_db_v3/Davos.html.

The AOD measurements for Innsbruck are available on request.

*Author contributions.* Jochen Wagner wrote the paper, performed most of the data analysis and made the figures. Alma Anna Ubele carried out part of the statistical analysis and updated the reference database. Verena Schenzinger has been responsible for the sunphotometer in Innsbruck and improved the AOD retrieval algorithm including cloud flagging. Axel Kreuter has operated the sunphotometer in Innsbruck for more than a decade. He was particularly involved in the planning of the paper and the figures.

*Competing interests.* The authors declare that they have no competing interests.

*Acknowledgements.* The authors acknowledge the Physikalisch-Meteorologisches Observatorium Davos / World Radiation Center (pmd/wrc) and AERONET-Europe/ACTRIS for long term operation and calibration and maintenance services of the CIMEL sunphotometer in Davos.

     This work was supported by the Medical University of Innsbruck.

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

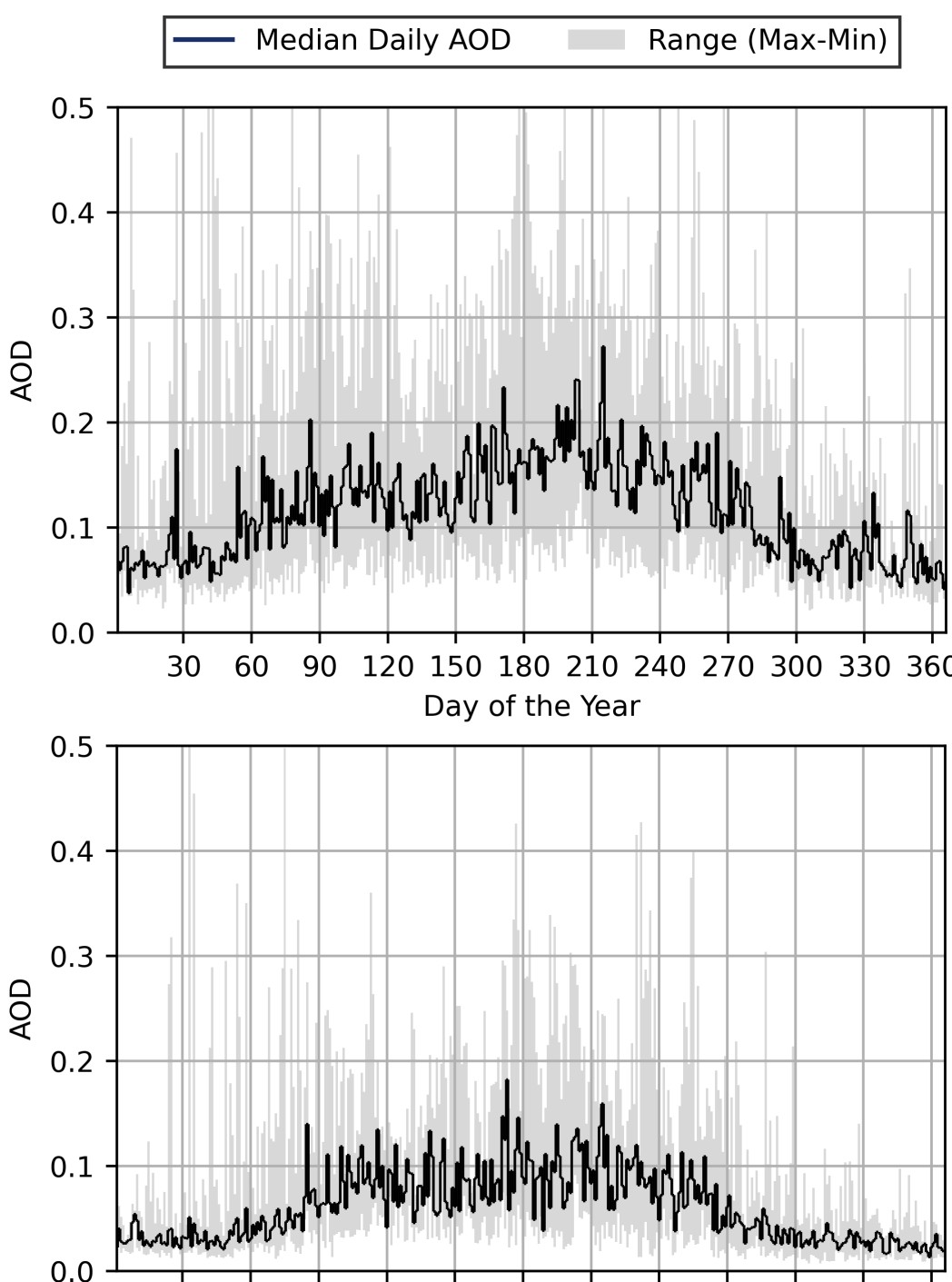

**Figure 3.** Innsbruck (top) and Davos daily 17 years AOD climatology. The median daily AOD is shown (black line) together with the min-max range (grey background).

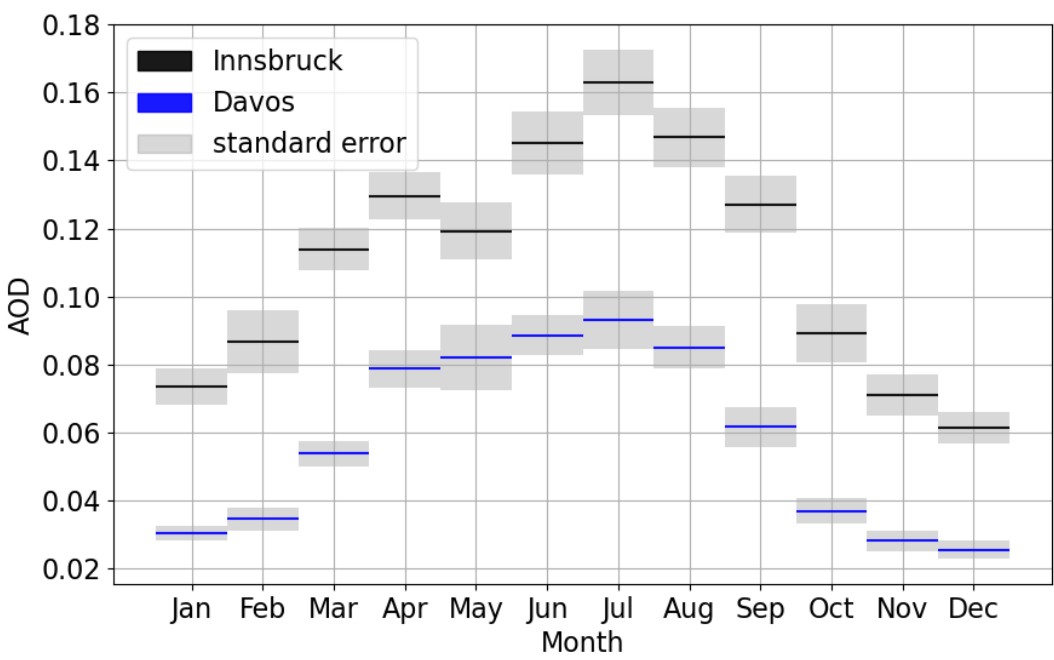

**Figure 4.** Monthly AOD 17 years climatology with standard errors for Innsbruck and Davos

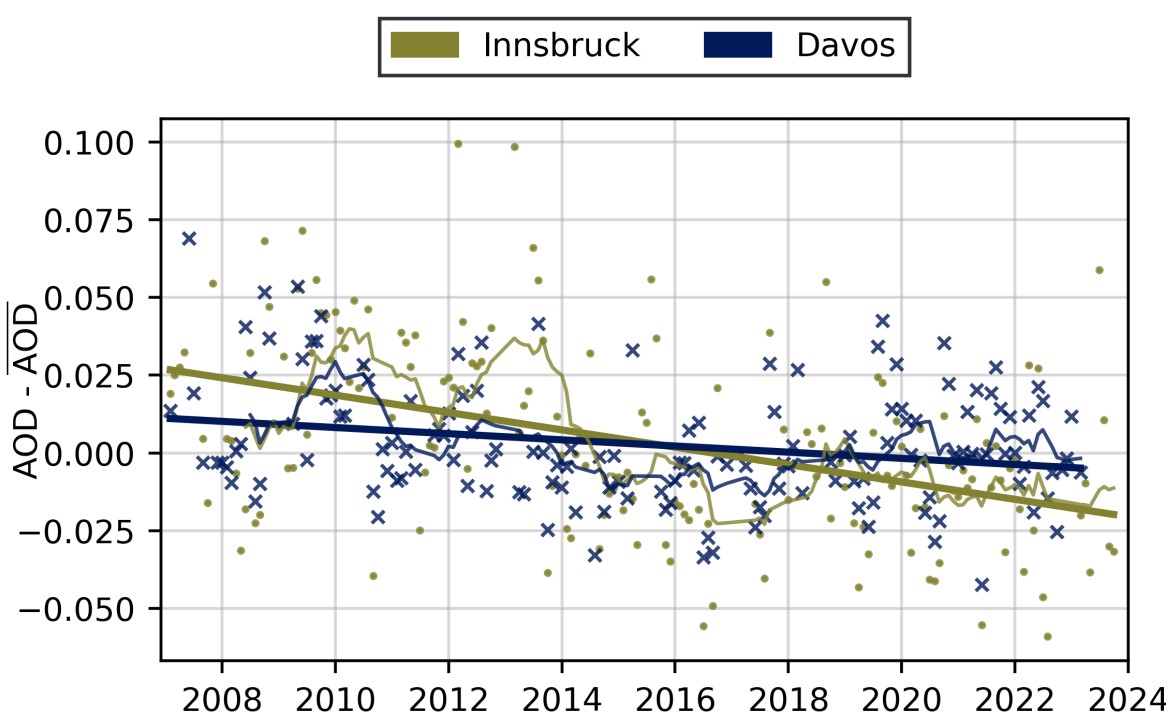

**Figure 5.** Deseasonalized monthly AOD for Innsbruck (greenish circles) and Davos (bluish crosses). The 12 month running mean (thin lines; Innsbruck - greenish, Davos - bluish) and the respective linear trends (thick lines)