# Peer review of "Extended Aerosol Optical Depth (AOD) time series analysis in an Alpine Valley: A Comparative Study from 2007 to 2023"

_Aerosol Research, 2023_

## Author Comment (AC1)

Review for "Extended Aerosol Optical Depth (AOD) time series analysis in an Alpine Valley: A Comparative Study from 2007 to 2023" by Wagner et al. (discussion 30 Jan 2024)

Valuable results in a well written paper. Very few mountain sites with such a long records of AOD. Methodology, clear and simple. Data at both sites (Innsbruck and Davos) have sufficient statistics and can be comparable. Already for this comparative aspect and the trends obtained for the AOD variability - it is worth publishing.

However, I'd like authors to rethink following:

Trends as both sites are provide as linear fit, maybe non-linear would be better? Why should one expect the linearly declining trend for the entre period? From the point of view of the low-elevation sites (AOD dominated by boundary layer aerosols) one maybe could expect that with the dimming (improving air quality in last decades) we get less AOD.  For the mountain site the AOD is related rather to the long-term aerosol transport, which at both sites can be expected significant, if not completely dominating. I feel this could be more properly addressed. I would less focus of the "remarkably similar trends" but try to explain better the differences and similarities  in Fig.8 in a way to explain why it is so.

*Thank you for your insightful observations regarding the use of linear versus non-linear models to analyze AOD trends at the two sites. The decision to employ linear regression in our analysis was guided by several considerations, primarily the length of the dataset and the preliminary nature of observed trends.*

*As noted, a typical climatology study often relies on datasets spanning at least 30 years to robustly characterize and interpret atmospheric trends. Given that our dataset covers approximately 17 years, it indeed presents limitations for a comprehensive climatological analysis. Within this shorter timeframe, linear regression provides a straightforward initial approach for identifying basic trends and patterns in the data, acknowledging that these results are preliminary and might change with the inclusion of more data over time.*

*Furthermore, the linear trend approach was selected due to its simplicity and transparency in interpretation, which is suitable for establishing a baseline understanding of the AOD dynamics at each site. However, we agree that the aerosol optical depth trends could be influenced by complex factors that a simple linear model might not fully capture.*

*At the low-elevation site in Innsbruck, AOD is significantly influenced by boundary layer aerosols, which have been shown to decrease in response to improved air quality measures over recent decades. This could suggest a non-linear response of AOD to ongoing environmental policy and technological changes. At the high-elevation site in Davos, long-range aerosol transport plays a more prominent role, potentially leading to different trends that could also be non-linear, influenced by changes in global aerosol emissions and atmospheric circulation patterns.*

*Addressing your point on the similarities and differences between the sites, it is evident that a more nuanced approach might better elucidate the distinct processes at each site. In future analyses, as more data become available, employing non-linear models or segmented linear models could be more appropriate to capture the potential phases of AOD changes due to both local management practices and global environmental shifts.*

*In conclusion, while the current study employs linear regression due to dataset constraints and aims for initial trend assessment, we acknowledge the need for more complex models to fully understand the temporal dynamics of AOD. Future work will aim to integrate longer time series and apply more sophisticated statistical techniques to better represent and understand the underlying processes affecting AOD at both sites.*

Limitations of the study in terms of not being ably to estimate of which % of AOD load is form boundary layer aerosols and which form free troposphere is not discussed. Taking into account that the Davos rural site measures only in free-troposphere (can this be assumed?), the Innsbruck urban site has a strong contribution form boundary layer aerosol. So are they comparable and to what extend. Are the similarities at both sites due only to high-tropospheric aerosol? Taking into account the latter site being one of the ACTRIS sites, it would be good to mention that continuous lidar observations of aerosol extinction profiles could help in such distinction.

*In terms of the aerosol source and distribution, it is plausible to consider that the Davos site, located at a high altitude, predominantly measures aerosols in the free troposphere. This assumption is based on its elevation and remote setting, which generally limits the influence of local boundary layer sources. Conversely, Innsbruck, situated in an urban valley, is significantly impacted by local emissions and boundary layer aerosols, which contribute to its AOD measurements. The similarities observed in the AOD trends at both sites may indeed be influenced more by high-tropospheric aerosols. However, we cannot investigate this hypothesis using the presented AOD-timeseries. Continuous lidar observations of aerosol extinction profiles could provide a clearer distinction between boundary layer and free tropospheric aerosols. Such measurements, which are part of the ACTRIS (Aerosol, Clouds, and Trace gases Research InfraStructure Network) activities at some sites, allow for the vertical profiling of aerosols and could significantly enhance the understanding of their spatial distribution and temporal dynamics. We are working on this topic in an ongoing project.*

*We have added this section to the results section:*

Continuous lidar observations of aerosol extinction profiles could provide a clearer distinction between boundary layer and free tropospheric aerosols.

*And change a sentences in the conclusions and outlook section:*

Further investigations taking local emissions and land use changes into account are worthwhile.
-→
For a better understanding of the aerosol behaviour, it is essential to distinguish between boundary layer aerosols and aerosols in the free troposphere and additionally investigations on local emissions and land use changes are worthwhile.

minor/technical comments:

Please check reference Tiw 2023, line 13

*corrected*

Fig.1 and Fig.2 Caption - pls check denoted colors are not in the figure

*Corrected, after updating the figures we did not update the caption – sorry for our clumsiness*

Longer data gaps occur at both sites de to device failures or calibration - can you quantify (e.g. 24 days due to X and 45 days due to y) to betetr assess on the instrument reliability?

We added two horizontal colorbars for a better visual detection of the data gaps. Our approach in Innsbruck has been to categorize these gaps into broader categories reflecting the primary causes, such as device failures, calibration, maintenance periods and measurement campaigns at a different location. In Innsbruck, we estimate that significant data gaps due to notable hardware and software issues encompass approximately 50% of the total measurement period, with the remainder largely attributed to planned maintenance and operational transitions at a different position (measurement campaigns.

Fig.5 the black dots are no visible

There are no black dots – we decided to omit the black dots for a clearer visualization, but again forgot to update the caption

line 64 and 76 - same info, pls avoid repetitions (please check also elsewhere)

We omit the sentences: "The average AOD in Davos (0.054) is about half as high as in Innsbruck (0.115)." and "This type of display is particularly suitable for daily data monitoring if the current data is also displayed in the graph."

It is a comparative study, so it would be better to plot Figs.1 and 2 as one figure with two panels one above other. Similar for e.g. Figs. 5 and 6, I would plot one next to other. This way you will reduce the length of paper and also ease reader life.

Thanks for the suggestion. We combined figure 1 and 2 and figure 5 and 6 accordingly.

Also, the most important result is the trend in Fig.8 but you have too many figures and this message gets lost.

By reducing the number of figure, we hope that the main message – the declining trends are – is now more visible

---

## Author Comment (AC2)

In this study, the authors compared the aerosol optical depth (AOD) time series between two alpine valleys from 2007 to 2023. The paper is well written. The methods used and the results obtained are reasonably well documented. The long data series is a treasure in this field of aerosol research. I therefore recommend the manuscript for publication in the journal Aerosol Research.

The authors, however, should consider the following questions and recommendations when preparing the final draft. (In specific cases I refer to the page and line numbers of the draft I received.)

A more precise description of the two sites would be important because many readers do not know anything about the two sites. Characterize the site, such as type of town, e.g. industrial area, cultural center, population, etc. Furthermore, please provide detailed information on the meteorological conditions.

*We have added this section to the introduction*:

*Innsbruck, situated in the broad Inn Valley, is a prominent cultural and academic center in western Austria with obout 132,000 residents. The city's geographical position in a large valley facilitates unique meteorological conditions, characterized by pronounced seasonal variations. Typical weather patterns include relatively dry winters and wetter summers, with occasional föhn winds influencing both temperature and precipitation levels. Davos, on the other hand, is a high-altitude town located in the Swiss Alps. It has a smaller population of about 11,000, which can swell significantly during tourist seasons. Davos experiences a subarctic climate, which includes long, snow-rich winters and cool summers. The meteorological setup in Davos leads to a distinct aerosol composition primarily influenced by tourism-related activities and seasonal sports events, contrasting with Innsbruck's more urban aerosol sources from vehicular traffic and industrial emissions. Both sites, therefore, offer contrasting environments for the study of aerosols, significantly enhancing the comparative analysis of long-term AOD trends.*

Page 2 Figure 1 and page 3 Figure 2: Both sites are missing data around 2013. Please explain.

*The data gap in Innsbruck range from October 20212 to January 2013. The data gap in Davos range from November 2012 to February 2013. The sun photometer from Innsbruck was in Davos for maintenance and calibration at this time. We added two horizontal colorbars for a better visual detection of the data gaps.*

The markers in Figure 1 and 2 are difficult to understand. Please correct.

*We combined figure 1 and 2 into one figure. The time series of Innsbruck and Davos are now better comparible and the markers are correctly explained in the caption.*

Page 3, Line 47: The monthly data is 73.2 % in the text while 83.2% in Table 1. Please correct it.

*The values in table 1 are correct., We changed the text (line 47) accordingly.*

In Figure 8 the markers cannot be followed. Which is the black dot and which is the blue one? Please clear them.

*We apologize for the error. After updating the figures we did not update the caption. The content of the ifigure should now be clear to the reader with the new caption:* "Deseasonalized monthly AOD for Innsbruck (greenish circles) and Davos (bluish crosses). The 12 month running mean (thin lines; Innsbruck - greenish, Davos - bluish) and the respective linear trends (thick lines)"

The authors use the term correlation several times in the text. In these cases I would expect correlation values with significance levels. Please correct them.

We calculated p-value and Person correlation coefficient (r) for all trends. The trend is considered significant if the conditions $p<0.05$ and $|r|<0.6$ are fulfilled.

---

## Author Comment (AC3)

Review for "***Extended Aerosol Optical Depth (AOD) time series analysis in an Alpine Valley: A Comparative Study from 2007 to 2023***" submitted to Aerosol Research by Wagner et al.

**Synopsis:**

This manuscript which analyses two long-term datasets from Aerosol Optical Depth observations in the Alps is a valuable contribution to the assessment of aerosol trends in mountainous areas.

**General comments:**

- The manuscripts lacks a description of the dataset concerning measurement uncertainties, especially concerning long-term stability of the observations and possible biases.
  Regular calibrations of the PFR in Innsbruck, conducted in Davos, have shown remarkably stable calibration coefficients for the 501 nm channel over the past 17 years, with relative changes ranging from -0.5% to +0.7%.

- I suppose that the AERONET sun photometer in Davos has been regularly calibrated according to AERONET standards. There is no information about the instrument in Innsbruck, concerning instrument type and associated uncertainties. Please provide information about the uncertainties of both instruments and a summary of the calibrations performed during the 17-year period.

  We added this paragraph:

  *Precision Filter Radiometers (PFRs) are engineered to assess background aerosol conditions and have participated in sun photometer intercomparisons, like the CIMEL devices used in the global AERONET network to ensure data quality assurance. The discrepancies between PFRs and CIMEL devices used in the global AERONET network (Holben, 2001), consistently fall within a +/- 0.01 AOD range. The Innsbruck PFR performed even better during the intercomparison campaign in Davos in October 2021 (Kazadzis, 2023). Long-term analyses confirm the excellent traceability of AERONET AOD measurements to the World AOD standard at 500 nm (Cuevas, 2019). Furthermore, regular calibrations of the PFR in Innsbruck, conducted in Davos, have shown remarkably stable calibration coefficients for the 501 nm channel over the past 17 years, with relative changes ranging from -0.5% to +0.7%.*

- Please provide a (very brief) introduction into how AOD is defined and about the general measurement principle of sun photometers, as there might be readers who are not so familiar with these topics.

  *This paragraph was added in the introduction chapter:*

  It quantifies the cumulative effect of aerosol scattering and absorption along the path of sunlight through the atmosphere. AOD is unitless and provides an indication of atmospheric clarity,essential for climatological and environmental research. The

primary method for determining AOD is through the use of sun photometers, which measure the direct solar irradiance reaching the Earth's surface. The basic principle behind these measurements is the Lambert-Beer law, a fundamental equation that relates the intensity of light to the properties of the material through which it is passing.

$$I = I_0(R) \cdot e^{2_0 -\tau(\lambda) \cdot m} \quad (1)$$
with

– I is the observed intensity of sunlight after passing through the atmosphere
– $I_0(R)$ is the original intensity of sunlight before entering the Earth's atmosphere, dependent on the sun-earth distance R
– $\tau(\lambda)$ is the optical depth at wavelength $\lambda$, which includes contributions from aerosols, gases, and other atmospheric constituents
– m is the optical air mass, a factor that accounts for the path length through the atmosphere, which depends on the solar zenith angle (m ~ cos(sza))
A detailed description of the retrieval of AOD from sunphotometer measurements in Innsbruck is given in Wuttke et al.
(2012) and in Sinyuk et al. (2020) for the AERONET AOD retrieval respectively.

- As you have data only from about 80% of the months: Are these missing months equally distributed over the year? And what is the main reason for missing months? Are the gaps caused by instrument failures or calibration periods? Because I could imagine that you get nearly every month 5 cloud-free days where you can derive AOD. Please comment on that!

  *In addressing the data availability concerns raised, it is important to clarify the distribution and causes of the missing months in our AOD dataset for both Innsbruck and Davos. If the sun photometer runs faultlessly from sunrise to sunset, it is indeed plausible to expect at least five cloud-free days per month for AOD data collection.* The missing data, accounting for approximately 20% of the total dataset, are not entirely uniformly distributed throughout the year. Our analysis indicates that the gaps in the Innsbruck timeseries more prevalent during the winter months, primarily due to shorter daylight hours. The primary reasons for data gaps at both stations are twofold: instrument calibration and failures. Calibration periods are scheduled routinely to ensure the accuracy and reliability of our measurements but result in temporary interruption of data collection.

- I do not see any negative AOD trend after 2014 anymore (Fig. 8). Do you have an explanation for the decline until 2013, but no more change afterwards? Are the trends the same if you separate into winter and summer seasons?

  Thank you for your observations regarding the AOD trends. We also notice the apparent shift in trends post-2014, as mentioned. We see a similar pattern, when analyzing the summer and winter seasons separately. However, the time series is not yet long enough to be able to draw solid conclusions about the trend from 2013 onwards. Therefore, we did not start a discussion to this comment within the paper.

- What is the statistical significance of the trends? Please give some information on that!

  We calculated p-value and Person correlation coefficient (r) for all trends. The trend is considered significant if the conditions $p<0.05$ and $|r|<0.6$ are fulfilled.

**Minor comments:**

- Line 47 and Table 1: In the table you write for valid months: 83.2% and 79.4%, whereas in the text you give 73.3%/79.4%. Please check these values!

  *The values in table 1 are correct., We changed the text (line 47) accordingly.*

- Line 69: Do you think that there are more convective clouds "during the melting period"? I would rather say "after the melting period", as heating of the ground is inhibited as long as there is snow cover - at least in the higher parts of the Alps.

  *We added* "and after" *to the sentences. Of course, the heating of the ground is inhibited as long as there is a high albedo due to snow cover. However, a broken snow cover in spring guarantees sufficient warming and at the same time enough moisture for cloud formation.*

**Technical comments:**

- Line 8: typo in "latitude

  *corrected*"

- Figures 1,2: Please check the colors – I don't see the blue color of the individual measurements, for me the triangles for the monthly means look blue

  *There is no blue color, after updating the figures we did not update the caption – sorry for our clumsiness. The cation and the figures are now in line.*

- Figure 2 has another size than fig. 1: Please provide a larger version of fig. 2

  *We combined figure 1 and 2 into one figure. The time series of Innsbruck and Davos are now better comparible*